# Synthesis and Antiviral Activity of Novel 1,3,4-Thiadiazole Inhibitors of DDX3X

**DOI:** 10.3390/molecules24213988

**Published:** 2019-11-04

**Authors:** Annalaura Brai, Stefania Ronzini, Valentina Riva, Lorenzo Botta, Claudio Zamperini, Matteo Borgini, Claudia Immacolata Trivisani, Anna Garbelli, Carla Pennisi, Adele Boccuto, Francesco Saladini, Maurizio Zazzi, Giovanni Maga, Maurizio Botta

**Affiliations:** 1Dipartimento Biotecnologie, Chimica e Farmacia, Università degli Studi di Siena, Via A. Moro 2, I-53100 Siena, Italy; annalaura.brai@gmail.com (A.B.); ronzinis@gmail.com (S.R.); lor.botta83@gmail.com (L.B.); claudiozamperini@yahoo.it (C.Z.); borgini.m4@gmail.com (M.B.); c.trivisani@gmail.com (C.I.T.); 2Istituto di Genetica Molecolare, IGM-CNR “Luigi Luca Cavalli-Sforza”, Via Abbiategrasso 207, I-27100 Pavia, Italy; valentina.riva01@universitadipavia.it (V.R.); agarbelli@gmail.com (A.G.); carlapennisi@outlook.it (C.P.); 3Dipartimento di Biotecnologie Mediche, Università degli Studi di Siena, 53100 Siena, Italy; adele.boccuto@gmail.com (A.B.); france.saladini@gmail.com (F.S.); maurizio.zazzi@gmail.com (M.Z.); 4Biotechnology College of Science and Technology, Temple University, BioLife Science Building, Suite 333, 1900 North 12th Street, Philadelphia, PA 19122, USA

**Keywords:** DDX3X, HIV-1, host proteins, antivirals

## Abstract

The human ATPase/RNA helicase X-linked DEAD-box polypeptide 3 (DDX3X) emerged as a novel therapeutic target in the fight against both infectious diseases and cancer. Herein, a new family of DDX3X inhibitors was designed, synthesized, and tested for its inhibitory action on the ATPase activity of the enzyme. The potential use of the most promising derivatives it has been investigated by evaluating their anti-HIV-1 effects, revealing inhibitory activities in the low micromolar range. A preliminary ADME analysis demonstrated high metabolic stability and good aqueous solubility. The promising biological profile, together with the suitable in vitro pharmacokinetic properties, make these novel compounds a very good starting point for further development.

## 1. Introduction

Human ATPase/RNA helicase X-linked DEAD-box polypeptide 3 (DDX3X) belongs to the DEAD-box proteins, a large family of ATP-dependent RNA helicases that are involved in many aspects of RNA metabolism [1]. In 2004, Yedavalli et al. highlighted its fundamental role in HIV-1 life cycle as a shuttle protein that is able to export the viral RNA from the nucleus to the cytoplasm [2], subsequently Ishaq and coworkers [3] demonstrated that DDX3X knockdown suppressed HIV replication without inducing apoptosis. Besides HIV, DDX3X is known to be an essential host factor for other major human viral pathogens, such as Hepatitis B and C viruses [4,5,6], as well as for emerging viruses, such as Dengue virus (DENV), West-Nile virus (WNV), and Japanese Encephalitis Virus (JEV) [7,8].

DDX3X has multiple enzymatic activities (ATPase and RNA helicase) and functional domains that might be targeted by potential inhibitors [9,10]. Drug discovery activity in this area has been mostly oriented towards ATPase inhibitors, while only few efforts have been devoted to affect the helicase activity of DDX3X by targeting the RNA binding pocket with small molecules [11,12,13,14]. Figure 1 shows the structures of known DDX3X ATPase inhibitors. The first ATPase DDX3X inhibitors were discovered in 2008 [15]. Among them, we identified **FE15** through a virtual screening approach. This compound, characterized by a rhodanine scaffold, showed low micromolar potency of inhibition against the DDX3X ATPase activity and it was able to inhibit the HIV-1 replication in MT4 cells with an EC_50_ of 86.7 µM, without showing cytotoxicity. Similarly, the ring expanded nucleosides (RENs) were found able to inhibit the ATP dependent activity of DDX3X and suppress HIV-1 replication in T cells and monocyte-derived macrophages [16]. In 2011, through a hit optimization protocol, we identified the second generation of rhodanine DDX3X inhibitors endowed with an improved activity profile (as an example **FE109** showed a Ki of 0.2 µM) [17]. In parallel, some REN derivatives were synthesized, among them compound **NZ51** has been reported to inhibit the ATP dependent helicase activity of DDX3X, as well as the proliferation of cancer cells expressing high levels of DDX3X [18]. Raman and coworkers have recently reported compound **RK-33** containing the diimidazo[4,5-d;4′,5′-*f*] [1,3]diazepine ring. **RK**-**33** has been extensively studied as an anticancer compound, and its spectrum of activity includes different tumor cell lines derived from lung (A-549 and H-460), prostate (PC-3), breast (MCF-7 and MDA-MB-231), and ovarian (OVCAR-3) cancers [19,20,21]. At the same time, **ketorolac** salt has been identified as a novel DDX3X inhibitor able to affect the ATPase activity and endowed with anticancer activity against oral squamous cell carcinoma (OSCC) cell lines [22].

Herein, docking studies were employed to guide the design of new DDX3X inhibitors endowed with a thiadiazole nucleus. Being the rhodanines promiscuous binders with poor selectivity and also considered to be Pan Assay Interference Compounds (PAINS) [23], we pursued the idea of replacing the rhodanine moiety with a different ring maintaining profitable interactions within the ATP binding pocket of DDX3X. As a result, a new series of inhibitors of the ATPase activity of DDX3X was identified that showed good anti-HIV activity. Taken together, our results led to the identification of a new family of DDX3X inhibitors that can be used as a starting point to identify novel preclinical candidates to treat viral diseases that are caused by DDX3X dependent pathogens.

## 2. Results

### 2.1. Molecular Modeling

Docking simulations were performed on the known ATPase DDX3X inhibitors **RK33** and **ketorolac** in order to study the key molecular interactions on the basis of their inhibitory activity and the predicted binding modes were compared to those previously published for the active rhodanine derivative FE15 as well as with that of the crystallized ligand AMP (Figure 2). Calculations were performed according to the already published protocol [15].

An analysis of the interactions established by the studied DDX3X inhibitors within the ATP binding pocket was performed. Common interactions emerged from such investigation. In detail, the studied inhibitors established a hydrogen bond with Gln207 and a π-π interaction with Tyr200 that resulted in being crucial for their inhibitory activity. Furthermore, **FE15** and ketorolac made polar contacts with Gly227 and Gly229, mimicking the phosphate group of AMP. Ketorolac, being endowed with an acidic group, better reproduced the phosphate interactions and indeed additionally interacted with Thr231. **FE15** and **RK**-**33** both also occupied a region delineated by the amino acids Thr201 and Arg202. Taking into consideration all of the interactions made by the known active ligands, a new series of compounds was designed that contains all of the features identified as being crucial for the activity against DDX3X. The rhodanine group included in our previous hit compound, **FE15**, was discarded. Indeed, rhodanine derivatives are promiscuous binders that have been associated with multiple biological activities. Furthermore, rhodanines have been described as PAINS and highly problematic frequent hitters [23,24]. Viceversa, the thiadiazole ring, is an important framework in medicinal chemistry. Thiadiazole is a bioisostere of pyrimidine and oxadiazole and given the prevalence of pyrimidine in nature it is unsurprising that thiadiazoles exhibit significant therapeutic potential. Accordingly, a number of thiadiazole-containing drugs are currently on the market [25]. For the above reasons, we decided to replace the rhodanine nucleus of the hit compound **FE15** with a 1,3,4-thiadiazole ring.

Docking studies predicted good poses for the novel series of compounds within the ATP binding pocket of DDX3X. The binding mode of compound **23** (as representative of the novel series) revealed its ability to establish (Figure 3), similarly to the active inhibitors **RK33**, **ketorolac**, and **FE15**, hydrogen bond interactions with the crucial residues Gln207, Gly227, Gly229, Thr231, Thr201, and Arg202. Furthermore, its binding mode is stabilized by a π-π contact between the thiadiazole heterocycle and the side-chain of Tyr200. Starting from this result, a small library of 1,3,4-thiadiazole derivatives has been rationally designed and synthesized.

### 2.2. Chemistry

The synthesis of final compounds **23**–**28** and **29**–**32** first entailed the synthesis of the key intermediates **5**–**7** and **14**–**16**.

According to Scheme 1, **5**, **6**, and **7** were obtained by condensation of the thiosemicarbazide **4** with the opportune benzoic acid **1**–**3**. 

The synthesis of intermediates **14**, **15**, and **16** is depicted in Scheme 2. Aromatic aldehydes **8**–**10** reacted with thiosemicarbazide **4** to furnish thiosemicarbazones **11**–**13**. Subsequent cyclization in the presence of FeCl_3_ provides intermediates **14**–**16**.

Intermediates **5**–**7** and **14**–**16** were reacted with phtalimide to furnish compounds **17**–**22**. Subsequent hydrolysis in presence of LiOH provides final acids **23**–**28.** Esters **29**–**32** were synthesized by the reaction of acids **23**–**28** with the opportune alcohoxydes (Scheme 3). As depicted in Scheme 4 sulfonic acids **33** and **34** were produced by reacting between aminothiazoles **5** and **16** and 2-Sulfobenzoic acid cyclic anhydride. Finally, reduction of compounds **34** and **28** led to anilines **35** and **36** (Scheme 5).

### 2.3. Biological Evaluation

Derivatives **23**–**36** were next evaluated for their ability to inhibit the ATPase activity of DDX3X.

As reported in Table 1, compounds were characterized by Ki values in the low micromolar range against the ATPase binding site, confirming the validity of the reported approach. The phtalimido compound **17** was found to be completely inactive, probably due to its chemical instability. The para-methoxy derivative **23** and its corresponding phenol derivative **24** are endowed with the best affinity values (Ki) of 1.5 µM and 1.9 µM, respectively. The substitution of acidic function with ester slightly reduced the activities, being derivatives **29**, **30** and **31** respectively 12.5, 16.8 and 18.8 times less active than compound **23**. The introduction of electron withdrawing groups, such as bromine and nitro in ortho position, abolished (compound **25**) or reduced the activity (compound **28)**, despite even in this case the substitution of the carboxylic acid with a methyl ester (compound **32**) annulled enzymatic inhibition. The introduction of dimethylamino group in para position slightly decreased the activity in compound **27** (Ki = 11.9), while the introduction of amino group induced the total loss of inhibitory capability of compound **36**. Finally, the replacement of the carboxylic group with sulfonic acid was detrimental for the activity of compounds **33**, **34**, and **35**.

Compounds that were endowed of the best anti-enzymatic activity values were then essayed against HIV-1 in H9 cells. As reported in Table 1, esters derivatives **29** and **31** had promising antiviral activity values of 36 µM and 16 µM. However, compound **28** had some antiviral activity that can be attributable to its cytotoxic effect (SI = 1.7). The best result is represented by compound **24**, characterized by a promising antiviral activity of 3.9 µM without signs of cytotoxicity (CC_50_ = 125 µM, SI = 45). Strikingly, its p-methoxy derivative **23** was found to be inactive. The reasons are unknown and under investigation.

### 2.4. ADME Assays

Selected compounds were then analyzed in vitro for liver microsomal stability, membrane passive permeability, and aqueous solubility at pH 7.4.

Microsomal metabolic stability was evaluated by incubating each compound at 37 °C for 60 min. in phosphate buffer in presence of human liver microsomes. The reaction mixtures were then centrifuged, and the parent drug and metabolites were subsequently determined by LC-UV-MS. As reported in Table 2, compounds **23** and **24** had good metabolic stability, and only little percentage of the parent compound was metabolized into the dealkylated and into the oxidized derivatives. Contrarily, esters **29–31** revealed low stability, and they were rapidly converted into their corresponding carboxylic acids. In particular, compound **29** was rapidly converted into methoxy derivative **23** and phenol derivative **24**.

Kinetic solubility was calculated adding 1 mg of compound into 1 mL of water. After 24 h of stirring at 27 °C, the mixture was filtered and the quantity of solubilized compound determined by LC-MS-MS. As reported in Table 1, compounds **23**, **24**, and **29** showed very promising solubility values. By contrast, esters **30** and **31** due to their lipophilicity have values outside the recommended range (−2 < LogS < −6).

Finally, we evaluated passive membrane permeability (AppP) while using Parallel Artificial Membrane Permeability Assay (PAMPA). The quantity of compound able to diffuse through a semipermeable artificial membrane was calculated by LC-MS-MS. As reported in Table 1, acids **23** and **24** showed low AppP, in contrast esters derivatives **29**–**31** possessing very good permeability values, directly proportional to the length of their sidechains.

## 3. Experimental Section

### 3.1. General Procedures

The reagents were from Sigma-Aldrich (Milan, Italy), Alfa Aesar (Kandel, Germany) and Merck. Commercially available CH_2_Cl_2_ and MeOH were dried to remove calcium hydride or magnesium methoxide contaminants.

Anhydrous reactions were performed at positive pressure in dry N_2_ or argon atmosphere. TLC was carried out on Merck silica gel 60 F254 TLC plates. Flash chromatography was performed on Merck 60 silica gel, 23–400 mesh, columns.

For NMR spectra, a Bruker Avance DPX400 spectrometer (Bruker, Billerica, MA, USA) at 400 MHz for ^1^H-NMR or 100 MHz for ^13^C-NMR was used. Reported chemical shifts are relative to tetramethylsilane at 0.00 ppm. The abbreviations used for 1H patterns were: s = singlet, d = doublet, t = triplet, q = quartet, quint = quintet, sx = sextet, sept = septet, m = multiplet, br = broad signal, and br s = broad singlet.

For mass spectra (MS), an Agilent 1100 LC/MSD VL system (G1946C, Agilent Technologies, Palo Alto, CA, USA) with a 0.4 mL/min. flow rate was used in a binary solvent system 25 of 95:5 methyl alcohol/water. UV were monitored at 254 nm. Positive and negative mode scanning for mass spectra acquisition over the mass range was used.

#### 3.1.1. General Procedure for the Synthesis of Amino Thiazoles **5**–**7**

A mixture of the opportune substituted benzoic acid (6.57 mmol), thiosemicarbazide (9.85 mmol) and phosphorus oxychloride (5 mL) was stirred at 75 °C for 4 h under N_2_ atmosphere. After cooling to room temperature, water was added and the reaction mixture was further refluxed for 4 h. The mixture was cooled and basified to pH 8 by the addition of 1N NaOH_(aq)_ solution. The resulting mixture was then extracted three times with EtOAc and then washed with water. Organic layer was dried over Na_2_SO_4_ and evaporated under reduced pressure. The crude mass was recrystallized from EtOH. The compounds were obtained as light-yellow solids in a yield of 75–80%.

*5-(4-Methoxyphenyl)-1,3,4-thiadiazol-2-amine* (**5**): crystallization in EtOH. Yellow solid, yield 75%. ^1^H-NMR (400 MHz, Acetone): δ 7.73–7.71 (d, *J* = 8.0 Hz, 2H), 7.01–6.99 (d, *J* = 8.0 Hz, 2H), 6.56 (bs, 2H), 3.84 (s, 3H) ppm. LC-MS (ES): *m*/*z* 230 [M + Na]^+^.

*4-(5-Amino-1,3,4-thiadiazol-2-yl)phenol* (**6**): crystallization in EtOH. Yellow solid, yield 30%.^1^H-NMR (400 MHz, Acetone): δ 8.88 (bs, 1H), 7.65–7.64 (d, *J* = 5.2 Hz, 2H), 6.92–6.91 (d, *J* = 5.2 Hz, 2H), 6.57 (bs, 2H) ppm.

*5-(2-Bromophenyl)-1,3,4-thiadiazol-2-amine* (**7**): Purification eluent DCM/MeOH 98:2, yield 50%, yellow solid. ^1^H-NMR (400 MHz, CDCl_3_): δ 7.95–7.93 (d, *J* = 8.0 Hz, 1H), 7.68–7.66 (d, *J* = 8.0 Hz, 1H), 7.42–7.38 (t, *J* = 8.0 Hz, 1H), 7.32–7.30 (t, *J* = 8.0Hz, 1H), 7.26 (bs, 2H) ppm.

#### 3.1.2. General Procedure for the Synthesis of Thiosemicarbazones **11**–**13**

The opportune aromatic aldehyde **8**–**10** (40 mmol) was dissolved in warm EtOH was added to an aqueous solution of thiosemicarbazide (40 mmol) with continous stirring. After 1 h, the white solid tha formed was filtered off as pure **11**–**13**.

*(**2-(3,4-Dimethoxybenzylidene)hydrazine-1-carbothioamide* (**11**): crystallization in EtOH. Yellow solid, yield 60%. ^1^H-NMR (400 MHz, DMSO-*d*_6_): δ 11.20 (bs, 1H), 8.07 (s, 1H),7.88 (s, 1H), 7.43 (bs, 2H), 7.06–7.04 (d, *J* = 8.0 Hz, 1H), 6.88–6.86 (d, *J* = 8.0Hz, 1H), 3.73 (s, 3H), 3.70 (s, 3H) ppm.

*2-(4-(Dimethylamino)benzylidene)hydrazine-1-carbothioamide* (**12**): crystallization in EtOH, yellow solid, yield 50%, ^1^H-NMR (400 MHz, DMSO-*d*_6_): δ 12.3 (bs, 1H), 8.41 (s, 1H), 7.75 (bs, 2H), 7.48–7.46 (d, *J* = 8.0 Hz, 2H), 6.81–6.6.79 (d, *J* = 8.0 Hz, 1H), 3.02 (s, 6H) ppm.

#### 3.1.3. General Procedure for the Synthesis of Thiadiazoles **14**–**16**

Te opportune thiosemicarbazone (5 mmol) was suspended in EtOH, to this FeCl_3_·6H_2_O (20 mmol) was added and the reaction mixture was stirred under reflux for further 12 h. The reaction mixture was quenched with aqueous Na_2_S_2_O_3_, cooled to room temperature, poured into 30 mL water under stirring, and then extracted with CH_2_Cl_2_ (three times). The organic layer was dried over anhydrous Na_2_SO_4_. The organic solvent evaporated and the corresponding residue was purified by recrystallization from EtOH to afford the corresponding thiadiazoles **14**–**16**.

*5-(3,4-Dimethoxyphenyl)-1,3,4-thiadiazol-2-amine* (**14**): crystallization in EtOH. Yellow solid, yield 44%. ^1^H-NMR (400 MHz, DMSO-*d*_6_): δ 7.28 (s, 1H), 7.22 (bs, 2H),7.13–7.12 (d, *J* = 2.8 Hz, 1H), 6.65–6.64 (d, *J* = 2.8 Hz, 1H), 3.75 (s, 3H), 3.72 (s, 3H) ppm.

*5-(4-(Dimethylamino)phenyl)-1,3,4-thiadiazol-2-amine* (**15**): crystallization in EtOH. yellow solid, yield 50%, ^1^H-NMR (400 MHz, DMSO-*d*_6_): δ 7.72–7.70 (d, *J* = 8.0 Hz, 2H), 7.57 (bs, 2H), 6.90–6.88 (d, *J* = 8.0 Hz, 2H), 3.14 (s, 6H) ppm.

*5-(2-Nitrophenyl)-1,3,4-thiadiazol-2-amine* (**16**). To a suspension of **2** (100 mg, 0.44 mmol) in EtOH (10 mL), FeCl_3_*6H_2_O (482.1 mg, 1.78 mmol) in EtOH (10 mL) was added and the resulting mixture was heated at reflux for 12 h. The reaction mixture was diluted with water, alkalized by 10% NaOH solution, and then extracted with DCM. The organic layer was evaporated and the residue was recrystallized from Ethanol (60% yield). ^1^H NMR (DMSO-*d*_6_): δ 7.93 (d, *J* = 7.4 Hz, 1H), 7.74–7.64 (m, 3H), 7.53 (s, 2H) ppm. ^13^C-NMR (DMSO-*d*_6_): δ 170.4, 150.9, 148.7, 133.2, 131.7, 131.2, 124.7, 124.0. LC-MS(ES) *m*/*z* = 222.7 [M + H]^+^, *m*/*z* = 244.8 [M + Na]^+^.

#### 3.1.4. General Procedure for the Synthesis of Isoindoline-1,3-Diones **17**–**22**

To a solution of substituted-1,3,4-thiadiazol-2-amine (0.48 mmol) in CH_3_CN, phtalic anhydride (1.4 mmol) and triethylamine (1.05 mmol) were added, and the corresponding mixture was stirred at reflux overnight under N_2_ atmosphere. The product was filtered and washed with EtOAc to give a light yellow solid.

*2-(5-(4-Methoxyphenyl)-1,3,4-thiadiazol-2-yl)isoindoline-1,3-dione* (**17**): Yellow solid, yield 78%. ^1^H-NMR (400 MHz, DMSO-*d*_6_): δ 8.06–8.04 (m, 2H), 7.98–7.96 (m, 4H, 7.12–7.10 (d, *J* = 8.0 Hz, 2H), 3.84 (s, 3H). ^13^C-NMR (DMSO-*d*_6_): δ 167.05, 164.74, 162.25, 152.88, 136.03, 131.59, 129.67, 124.74, 122.41, 115.95, 115.43, 114.87 ppm.

*2-(5-(4-Hydroxyphenyl)-1,3,4-thiadiazol-2-yl)isoindoline-1,3-dione* (**18**): filtration, yellow solid, yield 80%. ^1^H-NMR (400 MHz, DMSO-*d*_6_): δ 10.23 (bs, 1H), 8.05–8.02 (m, 2H), 7.97–7.94 (m, 2H), 7.86-7.84 (d, *J* = 8.0 Hz, 2H), 6.93–6.91 (d, *J* = 8.0 Hz, 2H).

*2-(5-(2-Bromophenyl)-1,3,4-thiadiazol-2-yl)isoindoline-1,3-dione* (**19**): filtration, yellow solid, yield 45%. ^1^H-NMR (400 MHz, CDCl_3_): δ 8.21–8.19 (d, *J* = 8.0 Hz, 1H), 8.08–8.04 (m, 2H), 7.91–7.88 (m, 2H), 7.75–7.73 (d, *J* = 8.0 Hz, 1H), 7.49–7.46 (t, *J* = 7.2 Hz, 1H), 7.38–7.34 (t, *J* = 8.0 Hz, 1H), 7.26 (s, 1H).

*2-(5-(3,4-**D**imethoxyphenyl)-1,3,4-thiadiazol-2-yl)isoindoline-1,3-dione* (**20**): filtration, Yellow solid, yield 63%. ^1^H-NMR (400 MHz, DMSO-*d*_6_): δ 8.08 (s, 1H), 8.00–7.98 (d, *J* = 8 Hz, 1H), 7.91–7.89 (d, *J* = 8.0 Hz, 1H), 7.50 (s, 1H), 7.42–7.40 (d, *J* = 8.0 Hz, 1H), 7.07–7.05 (d, *J* = 8.0 Hz 1H), 3.81 (s, 3H), 3.78 (s, 3H).

*2-(5-(4-(Dimethylamino)phenyl)-1,3,4-thiadiazol-2-yl)isoindoline-1,3-dione* (**21**): filtration, Yellow solid, yield 45%. ^1^H-NMR (400 MHz, CDCl_3_): δ 8.06–8.08 (d, *J* = 8.0 Hz, 2H), 7.92–7.88 (m, 4H), 7.26–7.25 (d, *J* = 3.2 Hz, 1H), 6.92–6.91 (d, *J* = 3.2 Hz, 1H).

*2-(5-(2-Nitrophenyl)-1,3,4-thiadiazol-2-yl)isoindoline-1,3-dione* (**22**). Crystallization MeCN (81% yield, brown solid). ^1^H NMR (DMSO-*d*_6_): δ 8.14 (d, *J* = 7.3 Hz, 1H), 8.08–8.06 (m, 2H), 7.97–7.94 (m, 3H), 7.90–7.82 (m, 2H). LC-MS(ES) *m*/*z* = 352.9 [M + H]^+^, *m*/*z* = 375.0 [M + Na]^+^.

#### 3.1.5. General Procedure for the Synthesis of Acids **23**–**28**

Lithium hydroxide monohydrate (0.18 mmol) was added to a suspension of substituted-(1,3,4-thiadiazol-2-yl)isoindoline-1,3-dione (0.15 mmol) in a (1:1) mixture of H_2_O:THF. The reaction mixture was stirred at room temperature for 2 h, and was then acidified with HCl 1N and extracted three times with EtOAc. The organic layer was dried over anhydrous Na_2_SO_4_ and the solvent was removed at reduced pressure to provide the final product as white solid, which was used in the next step without further purification.

*2-((5-(4-Methoxyphenyl)-1,3,4-thiadiazol-2-yl)carbamoyl)benzoic acid* (**23**): crystallization MeCN, yield 67% ^1^H NMR (DMSO): δ 12.98 (s, 1H), 7.90–7.86 (m, 2H), 7.65–7.56 (m, 2H), 7.06–7.04 (m, 2H), 3.79 (s, 3H) ppm ^13^C-NMR (DMSO): δ 167.55, 167.01, 161.76, 161.09, 158.06, 135.97, 131.87, 130.38, 130.30, 129.76, 128.56, 128.52, 128.48, 128.17, 122.17, 114.78, 55.42 ppm LC-MS(ES) *m*/*z* = 352.9 [M + H]^+^, *m*/*z* = 354.0 [M − H]^−^.

*2-((5-(4-Hydroxyphenyl)-1,3,4-thiadiazol-2-yl)carbamoyl)benzoic acid* (**24**): crystallization in EtOH. white solid, yield 75%, ^1^H-NMR (400 MHz, DMSO-*d*_6_): δ 13.16 (bs, 1H), 10.04 (s, 1H), 7.93-7.91 (d, *J* = 8.0 Hz, 1H), 7.79–7.77 (d, *J* = 8.0 Hz, 2H), 7.69–7.60 (m, 3H), 6.90–6.88 (d, *J* = 8.0 Hz, 2H). ^13^C-NMR (100 MHz, DMSO-*d*_6_): δ 167.96, 167.49, 162.60, 160.17, 158.21, 136.50, 132.32, 130.82, 130.22, 129.10, 128.64, 121.70, 116.56 ppm. 

*2-((5-(2-Bromophenyl)-1,3,4-thiadiazol-2-yl)carbamoyl)benzoic acid* (**25**): Purification eluent DCM/MeOH 96:4, yield 50%, white solid. ^1^H-NMR (400 MHz, DMSO-*d*_6_): δ 13.40 (bs, 1H), 7.97–7.93 (t, *J* = 8.0 Hz, 2H), 7.85–7.83 (d, *J* = 8.0 Hz, 1H), 7.68–7.61 (m, 3H), 7.58–7.54 (t, *J* = 8.0 Hz, 1H), 7.49–7.45 (t, *J* = 8.0 Hz, 1H). ^13^C-NMR (100 MHz, DMSO-*d*_6_): δ 168.18, 167.56, 160.70, 159.82, 136.20, 134.32, 132.44, 132.30, 132.21, 131.60, 130.96, 130.35, 128.79, 128.75, 121.89 ppm.

*2-((5-(3,4-Dimethoxyphenyl)-1,3,4-thiadiazol-2-yl)carbamoyl)benzoic* acid (**26**): crystallization in EtOH. White solid, yield 60%. ^1^H-NMR (400 MHz, DMSO-*d*_6_): δ 7.87–7.85 (d, *J* = 8 Hz, 1H), 7.64–7.54 (m, 3H), 7.45–7.41 (m, 2H), 7.04–7.02 (d, *J* = 8 Hz, 1H), 3.8 (s, 3H), 3.76 (s, 3H). ^13^C-NMR (100 MHz, CDCl_3_): δ 168.06, 167.43, 162.44, 158.58, 151.36, 149.65, 136.54, 132.41, 130.85, 130.61, 130.18, 128.60, 123.31, 120.86, 112.57 ppm.

*2-((5-(4-(Dimethylamino)phenyl)-1,3,4-thiadiazol-2-yl)carbamoyl)benzoic acid* (**27**): crystallization in EtOH. white solid, yield 98%, ^1^H-NMR (400 MHz, DMSO-*d*_6_): δ 13.13 (bs, 1H), 7.92–7.90 (d, *J* = 8.0 Hz, 1H), 7.76–7.74 (d, *J* = 8.0Hz, 2H), 7.96–7.59 (m, 3H), 6.80–6.68 (d, *J* = 8.0Hz, 2H). ^13^C-NMR (100 MHz, DMSO-*d*_6_): δ 167.86, 167.50, 163.07, 152.15, 136.62, 132.32, 130.77, 130.18, 128.62, 128.49, 117.90, 112.52 ppm.

*2-((5-(2-Nitrophenyl)-1,3,4-thiadiazol-2-yl)carbamoyl)benzoic acid* (**28**): Crystallization MeCN (73% yield, pale yellow solid). ^1^H NMR (DMSO-*d*_6_): δ 8.06–8.00 (m, 3H), 7.90–7.74 (m, 3H), 7.59–7.51 (m, 2H). ^13^C-NMR (DMSO-*d*_6_): δ 170.5, 167.7, 156.6, 149.0, 138.0, 133.5, 133.2, 132.2, 131.7, 131.3, 130.5, 130.0, 124.9, 124.5. LC-MS(ES) *m*/*z* = 370.9 [M + H]^+^, *m*/*z* = 393.0 [M + Na]^+^, *m*/*z* = 368.9 [M − H]^−^.

#### 3.1.6. General Procedure for the Synthesis of Esters **29**–**32**

A mixture of substituted-(1,3,4-thiadiazol-2-yl)isoindoline-1,3-dione (0.29 mmol) and the opportune alcohoxide (0.29 mmol) was stirred at room temperature for 16 h under N_2_ atmosphere. Mixture was diluted with diluted CH_3_COOH, the residue was filtered and the solvent was evaporated at a reduced pressure. The product was purified via flash chromatography on silica gel (eluent CHCl_3_/EtOH, 98/2) to furnish a white solid product.

*Methyl 2-((5-(4-methoxyphenyl)-1,3,4-thiadiazol-2-yl)carbamoyl)benzoate* (**29**): Crystallization MeCN (53% yield, white solid).^1^H NMR (DMSO-*d*_6_): δ 8.04–8.02 (m, 2H), 7.96–7.92 (m, 2H), 7.10–7.07 (m, 2H), 3.81 (s, 3H) ppm ^13^C-NMR (DMSO-*d*_6_): δ 167.05, 164.74, 162.25, 152.88, 136.03, 131.59, 129.67, 124.74, 122.41, 115.95, 115.43, 114.87 ppm LC-MS(ES) *m*/*z* = 370.1 [M + H]^+^.

*Ethyl 2-((5-(4-methoxyphenyl)-1,3,4-thiadiazol-2-yl)carbamoyl)benzoate* (**30**): Purification eluent CHCl_3_/EtOH 98:2, yield 10%, white solid. ^1^H-NMR (400 MHz, CDCl_3_): δ 8.00–7.98 (d, *J* = 8.0 Hz, 1H), 7.84–7.82 (d, *J* = 8.0 Hz, 2H), 7.63–7.54 (m, 3H), 6.97–6.95 (d, *J* = 8.0 Hz, 2H), 4.29–4.23 (q, *J* = 7.2 Hz, 2H), 3.83 (s, 3H), 1.22–1.20 (t, *J* = 8.0 Hz, 3H). ^13^C-NMR (100 MHz, CDCl_3_): δ 167.48, 165.99, 164.74, 132.24, 130.68, 130.38, 129.43, 128.82, 127.85, 114.61, 61.82, 55.44, 13.87 ppm. LC-MS(ES) *m*/*z* = 384.2 [M + H]^+^.

*Butyl 2-((5-(4-methoxyphenyl)-1,3,4-thiadiazol-2-yl)carbamoyl)benzoate* (**31**): Purification eluent CHCl_3_/EtOH 98:2, crystallization in EtOH, yield 23%, white solid. ^1^H-NMR (400 MHz, CDCl_3_): δ 8.07–8.05 (d, *J* = 8 Hz, 1H), 7.77–7.75 (d, *J* = 8.0 Hz, 3H), 7.70–7.62 (m, 2H), 6.99–6.97 (d, *J* = 8 Hz, 2H), 4.23–4.19 (t, *J* = 6.8 Hz, 2H), 3.88 (s, 3H), 1.58–1.54 (q, *J* = 6.8 Hz, 2H), 1.33–1.28 (q, *J* = 7.6 Hz, 2H), 0.82–0.79 (t, *J* = 7.2 Hz, 3H). ^13^C-NMR (100 MHz, CDCl_3_): δ 167.44, 166.16, 162.39, 161.56, 159.59, 135.58, 132.19, 130.55, 130.39, 129.96, 128.60, 128.51, 123.11, 114.49, 65.57, 55.47, 53.43, 30.47, 19.15, 13.64 ppm. LC-MS(ES) *m*/*z* = 412.1 [M + H]^+^.

*Methyl 2-((5-(2-nitrophenyl)-1,3,4-thiadiazol-2-yl)carbamoyl)benzoate* (**32**): The crude was purified by flash chromatography using 99:1 DCM:MeOH (35% yield, white solid).^1^H NMR (Acetone-*d*_6_): δ 8.07–7.67 (m, 8H), 3.82 (s, 3H). ^13^C-NMR (Acetone-*d*_6_): δ 167.3, 166.0, 160.0, 157.7, 149.2, 136.2, 132.9, 132.3, 132.0, 131.4, 130.7, 129.9, 129.5, 128.3, 52.0 ppm. LC-MS(ES) *m*/*z* = 385.0 [M + H]^+^, *m*/*z* = 406.9 [M + Na]^+^.

#### 3.1.7. General Procedure for the Synthesis of Sulfonic Acids **33** and **34**

2-Sulfobenzoic acid cyclic anhydride (0.99 mmol) and TEA (0.54 mmol) were added to a stirred solution of the opportune aminothiazole (0.25 mmol) in MeCN (5 mL). The reaction mixture was heated at reflux for 90 min. Then, water was added to the mixture and acidified to pH 2 with 1N HCl. The resulting mixture was extracted with AcOEt, the organic layer was washed with brine, dried over Na_2_SO_4_, filtered, and evaporated in vacuo. The crude was purified by flash chromatography while using the opportune eluent.

*2-((5-(4-Methoxyphenyl)-1,3,4-thiadiazol-2-yl)carbamoyl)benzenesulfonic acid* (**33**): The crude was purified by flash chromatography on silica gel (from 4:6 PE:AcOEt to 100% AcOEt). (68% yield, white solid). 1H NMR (DMSO-*d*_6_): δ 8.05 (d, *J* = 7.9 Hz, 1H), 7.88 (d, *J* = 7.9 Hz, 2H), 7.74 (t, *J* = 8.0 Hz, 1H), 7.64–7.61 (m, 2H), 7.06 (d, *J* = 7.9 Hz, 2H), 3.87 (s, 3H) ppm. ^13^C-NMR (MeOD): δ 166.88, 163.42, 161.89, 158.65, 143.12, 131.83, 130.88, 130.20, 129.95, 128.52, 127.30, 122.90, 114.50, 54.77 ppm. LC-MS(ES) *m*/*z* = 390.1 [M − H]^−^.

*2-((5-(2-Nitrophenyl)-1,3,4-thiadiazol-2-yl)carbamoyl)benzenesulfonic acid* (**34**): The crude was purified by flash chromatography using from 4:6 PE:AcOEt to 100% AcOEt. (54% yield, yellow solid).^1^H NMR (DMSO-*d*_6_): δ 13.85 (s, 1H), 8.10 (d, *J* = 7.9 Hz, 1H),7.93 (d, *J* = 7.9 Hz, 2H), 7.89–7.78 (m, 3H), 7.65–7.55 (m, 2H). ^13^C-NMR (DMSO-*d*_6_): δ 166.6, 160.2, 157.6, 149.0, 154.3, 133.6, 132.5, 132.0, 131.9, 131.4, 130.2, 127.4, 125.1, 124.0. LC-MS(ES) *m*/*z* = 404.9 [M − H]^−^.

#### 3.1.8. General Procedure for the Synthesis of Compounds **35** and **36**

Iron dust (0.61 mmol) and NH4Cl (0.061 mmol) were added to a solution of the opportune nitrocompound (0.12 mmol) in a 3:1 mixture of EtOH/H_2_O (13 mL). The reaction mixture was stirred at reflux for 30 min. After that, the mixture was filtered through a Celite pad, concentrated in vacuo, and then purified by flash chromatography on silica gel whileusing the opportune eluent. 

*2-((5-(2-Aminophenyl)-1,3,4-thiadiazol-2-yl)carbamoyl)benzenesulfonic acid* (**35**): From EtOAc to 98:2 AcOEt:MeOH (76% yield, yellow solid). ^1^H NMR (DMSO-*d*_6_): δ 7.91 (d, *J* = 8.2 Hz, 1H), 7.80 (d, *J* = 8.4 Hz, 1H), 7.60–7.49 (m, 3H),7.91 (d, *J* = 8.2 Hz, 1H),7.15 (t, *J* = 7.6 Hz, 1H), 6.87–6.83 (m, 3H), 6.62 (t, *J* = 7.4 Hz, 1H). ^1^H NMR (DMSO-*d*_6_ + D_2_O): δ 7.91 (d, *J* = 8.2 Hz, 1H), 7.80 (d, *J* = 8.4 Hz, 1H), 7.60–7.49 (m, 3H), 7.91 (d, *J* = 8.2 Hz, 1H), 7.15 (t, *J* = 7.6 Hz, 1H), 6.83 (d, *J* = 8.3 Hz, 1H), 6.62 (t, *J* = 7.4 Hz, 1H). ^13^C-NMR (DMSO-*d*_6_): δ 166.5, 164.7, 157.0, 147.1, 144.8, 131.8, 131.4, 131.2, 130.5, 130.3, 127.4, 116.7, 116.5, 111.6. LC-MS(ES) *m*/*z* = 376.9 [M + H]^+^, *m*/*z* = 374.9 [M − H]^−^.

*2-((5-(2-Aminophenyl)-1,3,4-thiadiazol-2-yl)carbamoyl)benzoic acid* (**36**). The residue was purified by flash chromatography from 100% AcOEt to 95:5 AcOEt:MeOH (31% yield, yellow solid). ^1^H NMR (DMSO-*d*_6_): δ 7.86 (d, *J* = 7.4 Hz, 1H), 7.79 (d, *J* = 7.6 Hz, 1H), 7.56–7.44 (m, 3H), 7.14 (t, *J* = 7.6 Hz, 1H), 6.90-6.84(m, 3H), 6.63 (t, *J* = 7.6 Hz, 1H). ^1^H NMR (DMSO-*d*_6_ + D_2_O): δ 7.86 (d, *J* = 7.4 Hz, 1H), 7.79 (d, *J* = 7.6 Hz, 1H), 7.56–7.44 (m, 3H), 7.14 (t, *J* = 7.6 Hz, 1H), 6.82 (d, *J* = 9.2 Hz, 1H), 6.63 (t, *J* = 7.6 Hz, 1H). ^13^C-NMR (DMSO-*d*_6_): δ 169.3, 167.4, 164.3, 158.2, 147.4, 134.2, 131.4, 131.2, 131.0, 130.8, 130.5, 129.8, 116.6, 116.2, 111.9.LC-MS(ES) *m*/*z* = 339.0 [M − H]^−^.

### 3.2. Enzymatic Assays

#### 3.2.1. Protein Expression and Purification

Recombinant his-tagged human full length DDX3X was cloned into the *E. coli* expression vector pET-30a(+). Shuffle T7 *E. coli* cells were transformed with the plasmid and grown at 37 °C up to OD_600_ = 0.7. DDX3X expression was induced with 0.5 mM IPTG at 15 °C O/N. The cells were harvested by centrifugation, lysed, and the crude extract centrifuged at 100.000xg for 60 min. at 4 °C in a Beckman centrifuge before being loaded onto a FPLC Ni-NTA column (GE Healthcare). Column was equilibrated in Buffer A (50 mM Tris-HCl pH 8.0, 250 mM NaCl, 25 mM Imidazole, and 20% glycerol). After extensive washing in Buffer A, the column was eluted with a linear gradient in Buffer A from 25 mM to 250 mM Imidazole. Proteins in the eluted fractions were visualized on SDS-PAGE and then tested for the presence of DDX3X by Western blot with anti-DDX3X A300-475A*(BETHYL)* at 1:2000 dilution in 5% milk. Fractions containing the purest DDX3X protein were pooled and stored at −80 °C.

#### 3.2.2. ATPase Assay

The ATPase assay was carried out by using the commercial kit Promega (Milan, Italy), ADP-Glo^TM^ Kinase Assay. Reaction was performed in 30 mM TrisHCl pH 8, 9 mM MgCl_2_, 0.05 mg/mL BSA, 50 μM ATP, and 4 μM of recombinant DDX3X. Reaction was performed following the ADP-Glo^TM^ Kinase Assay Protocol and luminescence was measured with MicroBeta TriLux (Perkin Elmer, Milan, Italy).

### 3.3. Antiviral Assay

The antiviral activity was evaluated by measuring the half maximal inhibitory concentration (IC50) values against the HIV-1 wild-type reference strain NL4-3 in a TZM-bl cell line based phenotypic assay, named BiCycle Assay [26]. The method includes a first round of infection of human T cell lymphoma derived clone H9 at multiplicity of infection of 0.03 in the presence of serial dilution of compounds in a 96-well plate. After 72 h, 50 microliters of supernatants from each well were used to infect the TZM-bl cell line, which allow the quantitative analysis of HIV infection by measuring the expression of the luciferase gene integrated in the genome of the cells under the control of HIV-1 LTR promoter. The contribution of DDX3X expression in TZM-bl cells does not contribute in a relevant way to the determination of viral activity, as the reporter TZM-bl cells are minimally exposed to investigational compounds.

After 48 h, dose-response curves were generated by measuring reporter gene expression in each well by using the Bright-Glo Luciferase Assay (Promega) through the GloMax Discovery reader (Promega). Relative luminescence units measured in each well were elaborated with the GraphPad Software v.6.0 to calculate IC_50_ values. All the viruses and cell lines were obtained through the NIH AIDS Reagent Program (www.aidsreagent.org).

### 3.4. Cytotoxicity Assay

Cytotoxicity in H9 cells was determined by using the CellTiter-Glo 2.0 assay (Promega). H9 cells were seeded at 40,000 cells/well in duplicate in the presence of serial two-fold dilutions of compounds (range 200–1.56 μM) and incubated for 72 h. Cell viability was calculated by measuring cellular ATP as a marker of metabolically active cells through a luciferase based chemical reaction and expressed as the concentration that reduce cell viability by 50%.

### 3.5. ADME Assay

*Chemicals.* All solvents and reagents were from Sigma-Aldrich Srl (Milan, Italy). Dodecane was purchased from Fluka (Milan, Italy). Pooled Male Donors 20 mg/mL HLM were from BD Gentest-Biosciences (San Jose, California). Milli-Q quality water (Millipore, Milford, MA, USA) was used. Hydrophobic filter plates (MultiScreen-IP, Clear Plates, 0.45 μm diameter pore size), 96-well microplates, and 96-well UV-transparent microplates were obtained from Millipore (Bedford, MA, USA).

#### 3.5.1. Parallel Artificial Membrane Permeability Assay (PAMPA)

Donor solution (0.5 mM) was prepared by diluting 1 mM dimethylsulfoxide (DMSO) compound stock solution while using phosphate buffer (pH 7.4, 0.025 M). Filters were coated with 5 μL of a 1% (w/v) dodecane solution of phosphatidylcholine for intestinal permeability. Donor solution (150 μL) was added to each well of the filter plate. To each well of the acceptor plate was added 300 μL of solution (50% DMSO in phosphate buffer). All of the compounds were tested in three different plates on different days. The sandwich was incubated for 5 h at room temperature under gentle shaking. After the incubation time, the plates were separated, and samples were taken from both receiver and donor sides and analyzed while using LC with UV detection at 280 nm. LC analysis was performed with a PerkinElmer (series 200) instrument that was equipped with an UV detector (PerkinElmer 785A, UV/vis Detector). Chromatographic separation was conducted using a Polaris C18 column (150–4.6 mm, 5 μm particle size) at a flow rate of 0.8 mL min^−1^ with a mobile phase composed of 60% ACN/40% H_2_O-formic acid 0.1% for all compounds. Permeability (P_app_) was calculated according to the following equation with some modification to obtain permeability values in cm/s,
(1)$$Papp=\frac{VDVA}{\left(VD+VA\right)At}-\ln\left(1-r\right)$$
where *V_A_* is the volume in the acceptor well, *V_D_* is the volume in the donor well (cm^3^), *A* is the “effective area” of the membrane (cm^2^), *t* is the incubation time (s), and *r* the ratio between drug concentration in the acceptor and equilibrium concentration of the drug in the total volume(*V_D_* + *V_A_*).

Drug concentration is estimated by using the peak area integration. Membrane retentions (%) were calculated according to the following equation:(2)$$\%MR=\frac{[r-(D+A)]100}{Eq}$$
where *r* is the ratio between drug concentration in the acceptor and equilibrium concentration, *D*, *A*, and *Eq* represented drug concentration in the donor, acceptor, and equilibrium solution, respectively.

#### 3.5.2. Water Solubility Assay

The appropriate (1 mg) was added to 1 mL of water. The sample was shaken in a shaker bath at room temperature for 24–36 h. The suspensions were filtered through a 0.45 μm nylon filter (Acrodisc, Sigma Aldrich, Milan, Italy) and the solubilized compound determined by LC-MS-MS assay. The determination was performed in triplicate. For the quantification, an LC-MS system consisting of a Varian apparatus (Varian Inc., Palo Alto, CA, USA), including a vacuum solvent degassing unit, two pumps (212-LC), a Triple Quadrupole MSD (Mod. 320-LC) mass spectrometer with ES interface, and Varian MS Workstation System Control Vers. 6.9 software, was used. Chromatographic separation was obtained while using a 3 μm particle size Pursuit C18 column (50 × 2.0 mm) (Varian). Column was equilibrated in Buffer B (aqueous solution of 0.1% formic acid) and eluted with a linear increase from 0% to 70% gradient of Buffer A (ACN) for 10 min. and subsequent increase up to 98% for additional 5 min. The instrument operated in positive mode and the following parameters were used: injection volume 5 μL, flow rate 0.3 mL/min, detector 1850 V, drying gas pressure 25.0 psi, desolvation temperature 300.0 °C, nebulizing gas 45.0 psi, needle 5000 V, and shield 600 V. Nebulizer gas and drying gas was Nitrogen. Argon was the collision gas at a pressure of 1.8 mTorr in the collision cell. Single compound quantification was performed in comparison with standard calibration curves in methanol.

#### 3.5.3. Microsomal Stability Assay

Each compound was incubated at a final concentration of 50 μM in a final volume of 0.5 mL in 2% (final solution) DMSO for 60 min. at 37 °C under the following conditions: 125 mM phosphate buffer (pH 7.4), 5 μL of human liver microsomal protein (0.2 mg/mL) and NADPH-generating system. The reactions were stopped by adding 1.0 mL of acetonitrile and cooled in ice. Parent drug and metabolites in the reaction mixtures were determined after centrifugation of the samples by LC-UV-MS.

An Agilent 1100 LC/MSD VL system (G1946C) (Agilent Technologies, Palo Alto, CA, USA) was used for chromatographic analysis. A vacuum solvent degassing unit constituted the instrument, a binary high-pressure gradient pump, an 1100 series UV detector, and an 1100 MSD model VL benchtop mass spectrometer. A Varian Polaris C18-A column (150–4.6 mm, 5 μm particle size) was used under the following conditions: eluent A: ACN; eluent B: aqueous solution of formic acid (0.1%); linear gradient 2–70% of eluent A for 12 min, then increased to 98% up to 20 min.; flow rate 0.8 mL min.-1; injection volume 20 μL. The LC-ESI-MS determination was performed with Agilent 1100 series mass spectra detection (MSD) single-quadrupole instrument in the positive ion mode and an orthogonal spray API-ES (Agilent Technologies, Palo Alto, CA, USA). Nebulizing and drying gas was Nitrogen. Pressure of the nebulizing gas 40 psi, flow of the drying gas 9 L/min, capillary voltage 3000 V, fragmentor voltage 70 V, and vaporization temperature 350 °C. UV were monitored at 280 nm. Scan range for spectra acquisition was *m*/*z* 100–1500 while using a step size of 0.1 u. The percentage of compound not metabolized was determined by comparison with reference solutions.

## 4. Discussion

In the present paper, 15 novel inhibitors of the ATPase activity of the human helicase DDX3X were designed, synthesized, and biologically characterized. The compounds were designed starting from hits previously published, by replacing their rohdanine moiety, already reported in the list of PAINS, with the 1,3,4-thiadiazole ring. As a result, nine compounds showed promising anti- enzymatic activities varying from 1.5 µM to 35 µM. Compounds that were endowed of the most interesting activities against the enzyme were then evaluated on HIV-1 infected H9 cells to discover potential antiviral drugs. The antiviral screening led to the identification of seven antiviral compounds endowed of activities ranging from 2.8 µM to 42 µM. Among them, compound **24** was characterized by the best anti-HIV-1 activity of 2.8 µM and by a promising selectivity index (CC_50_ = 125 µM, SI = 45). The ADME assays were finally performed to preliminary evaluate the pharmacokinetic parameters of compounds and to rationalize results. Acid derivatives showed very good metabolic stability, in contrast to their corresponding esters, which were rapidly metabolized into the acid precursors. The aqueous solubility was found very promising; in fact, the values of the most active compounds were below 6, as recommended for drug candidates. Finally, compounds were characterized by low but not limiting passive permeability. Taking into account ADME results, the antiviral activities of the esters **29** and **31** is probably due to their enhanced membrane permeability, followed by a cellular conversion into compound **24**. Caco-2 experiments will be performed in the due time to evaluate the involvement of active transport mechanisms. 

In conclusion, in the present work, we have successfully replaced the rhodanine moiety of compound **FE15** with 1,3,4-thiadiazole ring. This approach is useful in reducing the interferences in assays that are generally associated with PAINS. The most promising derivative, compound **24**, is characterized by good inhibition of the ATPase activity of DDX3X protein and by promising anti-HIV-1 activity in a reference virus-cell system without signs of in vitro cytotoxicity. These data, together with suitable in vitro PK properties, make inhibitor **24** a good starting point for further optimization.

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
