# Peer review of "Synthesis and Antiviral Activity of Novel 1,3,4-Thiadiazole Inhibitors of DDX3X"

_molecules, 2019, doi:10.3390/molecules24213988_

Round 1
Reviewer 1 Report
Overall, this is a well presented manuscriptdescribing the synthesis of novel inhibitor of DDX3X and its antiviral activity. Preliminary data is presented to indicate that DDX3X inhibitors with a thiadiazolenucleus exhibit good anti-viral activity. The docking studies and the synthesis of these new compounds are well justified. However, what is not clear is why inhibiting DDX3X does not result in cell death of TZM-bl cells. Also, what is expression levels of DDX3X in TZM-bl cells.
Author Response
We tank the Reviewer for the appreciation of our work and for the relevant question. As described in paragraph 3.1, the antiviral activity was performed in a cell based assay consisting in a first round of infection where H9 cells were infected with HIV-1 NL4-3 strain in presence of dilution of DDX3X inhibitors. Harboring the luciferase gene under the control of HIV-1 LTR, TZM-bl cells were only used to measure the amount of viral particles produced by H9 cells in presence of serial dilution of DDX3 inhibitors and minimally exposed to investigational compounds. According to this, the contribution of DDX3X expression in TZM-bl cells does not contribute in a relevant way to the determination of viral activity in the BiCycle Assay. We have inserted this consideration in the revised text (lines 410-412). For these reasons, we did not determine the expression levels of DDX3X in TZM cells. However, this cell line is a derivative of HeLa cells. We have indeed previously determined the mean intracellular concentration of DDX3X protein in HeLa cells, with the method described in Brai et al. J Med Chem. 2019 Mar 14;62(5):2333-2347. The DDX3X concentration was 124(+/-)24 nM.
Reviewer 2 Report
This manuscript describes the novel inhibitors of DDX3X and the antiviral activity.
The authors made a docking study of several known inhibitors and the ATP pocket of human DDX3 and designed a new scaffold having a thiadiazole in place of rhodamine. They systematically synthesized a series of thiadiazole derivatives and obtained promising compounds of good activity and ADME properties. Therefore, this manuscript can be accepted in Molecules after a minor revision considering the following comment.
Line 117, page 4, for “haldehyde” read “aldehyde”.
Line 146, page 7, the authors describes “compound 17 was found completely inactive, probably due to its chemical instability”. Are there any evidence showing the instability of 17, such as decomposition etc? The Reviewer suspects the phthalimide structure of 17 because compound 17 is a sole phthalimide derivative listed in Table 1. Other compounds have 2-carboxybenzamide structure that looks important in activity.
Author Response
We thank the Reviewer for the appreciation of our work and for the relevant questions. As for the stability of compound 17, we agree with the Reviewer that the chemical nature of phtalimide group could be responsible for its lack of activity, indeed we did not pursue that particular structure any further. We observed a tendency of the compound to precipitate in solution during the enzymatic assay, a well as upon storage but we did not characterize further the nature of the instablity since we focused on the active compounds.
The word aldehyde has been corrected, we apologize for the oversight.